# Precision Medicine and Childhood Asthma: A Guide for the Unwary

**DOI:** 10.3390/jpm12010082

**Published:** 2022-01-10

**Authors:** Mark L. Everard

**Affiliations:** Division of Child Health, Children’s Hospital, Faculty of Medicine, University of Western Australia, Perth, WA 6009, Australia; mark.everard@uwa.edu.au

**Keywords:** asthma, misdiagnosis, children, airway smooth muscle, loss of homeostasis

## Abstract

Many thousands of articles relating to asthma appear in medical and scientific journals each year, yet there is still no consensus as to how the condition should be defined. Some argue that the condition does not exist as an entity and that the term should be discarded. The key feature that distinguishes it from other respiratory diseases is that airway smooth muscles, which normally vary little in length, have lost their stable configuration and shorten excessively in response to a wide range of stimuli. The lungs’ and airways’ limited repertoire of responses results in patients with very different pathologies experiencing very similar symptoms and signs. In the absence of objective verification of airway smooth muscle (ASM) lability, over and underdiagnosis are all too common. Allergic inflammation can exacerbate symptoms but given that worldwide most asthmatics are not atopic, these are two discrete conditions. Comorbidities are common and are often responsible for symptoms attributed to asthma. Common amongst these are a chronic bacterial dysbiosis and dysfunctional breathing. For progress to be made in areas of therapy, diagnosis, monitoring and prevention, it is essential that a diagnosis of asthma is confirmed by objective tests and that all co-morbidities are accurately detailed.

## 1. Introduction

No disease was less understood by medical men, than asthma. Every difficulty of breathing, if fixed and continuous, was designated asthmatic; and the same indefinite application of the term still remains in vulgar use. This general application of the word caused it to be employed to denote a variety of morbid states of the lung, very different from one another.

A practical treatise on the principal diseases of the lungs, GH Weatherhead. 1837.

This article is aimed at assisting scientists, working in or considering working in the field of precision medicine as applied to children with asthma, in understanding the challenges and difficulties of working in an area where even the criteria for diagnosis are not agreed upon. In part, this is to ensure studies are well designed and useful data are generated; and in part to enable scientists to challenge clinicians as to the suitability of the subjects being studied. To subject individuals to procedures for research without the likelihood of robust data being generated is not only a waste of time and money, it is also unethical.

Discussing precision medicine as applied to paediatric asthma is a daunting prospect, given that clinicians cannot agree upon what constitutes asthma, have apparently come close to giving up on trying to define it and all too frequently misdiagnose it, with both over and underdiagnosis depressingly common [1]. As noted by Dr. Weatherhead in 1837, the label is widely used as a default diagnosis by clinicians who lack an insight into the subtleties of airway disease and the wide range of conditions that have similar symptoms and/or signs. Sadly, little has changed in the intervening 194 years and the same sentiments are as valid today as when he wrote them. Unfortunately, most physicians continue to rely on the well-known ‘elephant test’, believing that although it is difficult to describe, you know it when you see it. Unfortunately, in the case of asthma, this test is all too fallible, with studies reporting that 50% or more of those ‘diagnosed’ as having asthma do not have asthma. This sorry state of affairs is in large part due to a failure to seek objective evidence that airway smooth muscle homeostasis is compromised and that ASM shortening is a key component of the symptoms a patient is experiencing. It is important to recognise that ‘doctor-diagnosed asthma’ (that is a diagnosis reached without objective evidence of rapid changes in ASM length) is of little value to those seeking to explore the utility of the tools available to those interested in precision medicine [1,2,3,4].

Considering that asthma is often cited as being the commonest chronic disease of childhood, it is extraordinary that there is no universally agreed upon definition. The UK’s most recent SIGN/BTS national asthma guidelines [5] decline to even try and define the condition. They also suggest that there is no consistent gold-standard diagnostic criteria such that it is not possible to make unequivocal evidence-based recommendations on how to make a diagnosis of asthma [5]. Such a lack of precision would suggest that it is an area that is crying out for the power of modern technology and data processing. However, if one is not able to agree upon the nature of the condition and how it should be diagnosed, it is likely that conflicting results will be generated depending on the characteristics of the subjects included in a study. The failure to agree upon diagnostic criteria and a lack of understanding of the key components of the condition compromise most studies as it can be difficult to know if the subjects are indeed suffering from asthma and, perhaps more importantly, whether the results of any given study reflect critical components of ‘asthma’ or, as is more likely, features of co-existing conditions. While a number of clinicians have over the years argued that the term should be abandoned due to its lack of precision [6,7], this does not provide a satisfactory solution in the absence of an alternative approach to managing a common but widely misunderstood condition. For a condition that some claim does not exist, there are a remarkable number of publications devoted to it, with 7925 in the year 2021 to the 9th September, representing 31 publications every single day of the year! Moreover, a search for ‘asthma’ on PubMed generates over 200,000 returns. Not bad for something that some argue does not exist and for which it seems obtaining consensus regarding its fundamental nature appears to be beyond us. Others have argued that asthma, chronic obstructive pulmonary disease (COPD) and other diseases affecting the conducting airways of the lungs are simply different manifestations of similar genetic and environmental factors and propose that the term chronic non-specific lung disease (CNSLD) is more appropriate. This is referred to as the ‘Dutch hypothesis’ [8]. While the same patient can manifest a number of pathologies such as the smoking asthmatic with small airway obstruction (COPD) who has long-standing chronic bacterial bronchitis (chronic bacterial dysbiosis) and bronchiectasis (dilation of bronchi, not a disease), this does not support the idea that they are all linked by genetic factors and many patients do not have multiple respiratory comorbidities [9]. Others argue that rather than focusing on a single condition, we should consider that there is a spectrum of disorders that make up ‘asthma’ [10], though this is just another example of displacement activity undertaken by clinicians when challenged with difficult tasks.

## 2. Why Attempt to Apply Precision Medicine to a Condition That Does Not Exist?

If asthma does not exist, undertaking ‘asthma’ studies would be pointless. Asthma does exist but in order to focus on childhood asthma, one needs to understand the key component that defines asthma and separate this from processes that may exacerbate or mimic the effects of loss of ASM stability.

## 3. Key Components of Asthma

### Guidelines Are of Little Help

When ‘guidelines’ do attempt to define ‘asthma’, they generally refer to it as having two key features in airway ‘hyper-responsiveness’ and ‘airway inflammation’ [11,12]. They often directly attribute the hyper-responsiveness to inflammation as in the 1990 British guidelines, which state *‘As a result of inflammation the airways are hyperresponsive and will narrow easily in response to a wide range of stimuli’* [13]. As noted above, more recent iterations of the same guideline have abandoned any attempt to define asthma.

The characteristics of the characteristic ‘inflammation is seldom discussed in the guidelines though they often imply that atopic Th2 inflammatory responses with ‘eosiniophilic’ inflammation (in which the percentages of eosinophils in samples from the airways is >3% of observed white cells in blood they represent 1–6%) is typical. This is despite the fact that this type of inflammation cannot be demonstrated in the majority of asthmatics worldwide. More importantly, there are many who demonstrate elevated eosinophils in sputum or in the airways who are either asymptomatic or experience a troublesome cough but who do not have narrowing of their airways [14,15]. Critically, these observations imply that eosinophils and allergic inflammation are neither necessary nor sufficient to induce asthma [1]. More recently, guidelines have either abandoned attempting a definition or modified their position to state that inflammation is *usually* but not universally a feature, thus abandoning the previous certainties linking inflammation and hyper-responsiveness [5,11,12,15,16].

Airway inflammation is typically observed in all chronic airway conditions including allergic bronchitis, bacterial bronchitis (due to bacterial dysbiosis), smoking-related bronchitis, COPD, bronchiectasis with a bacterial bronchitis and emphysema. Hence, airway inflammation is not unique to asthma and no pathognomonic asthmatic inflammatory responses have been identified. Thus, any markers of inflammation noted in asthmatic subjects may be related to asthma; but more commonly, it is likely to be a related co-morbidity.

In contrast to eosinophils, neutrophils do not appear to release mediators that promote ASM shortening. The observation that atopic individuals who also happen to have asthma experience symptoms persuaded many to suggest a causal link rather than recognise it as an association in which asthmatic symptoms can be exacerbated by mediators released by cells such as eosinophils, basophils and mast cells.

Asthmatic subjects in whom there is no evidence of a chronic bronchitis, allergy or viral infection may still exhibit some evidence of inflammation on occasions given that it is likely that stress applied to the epithelial, ASM themselves and related cells due to ASM shortening will of itself drive release of cytokines that will induce an inflammatory reaction.

## 4. Asthma Is an Acquired Loss of Post-Natal ASM Homeostasis

What is unique to asthma is airway hyper-responsiveness, that is airways will narrow due to airway smooth muscle shortening in response to stimuli that will not cause significant airway narrowing in individuals without asthma. It is an acquired loss of post-natal ASM homeostasis which normally keeps airway tone constant presumably with oscillations around an ideal airway smooth muscle length [1,17]. The cause of this loss of homeostasis is unknown, which is not surprising given that the key components of control are still not understood. Helping to define this may be ‘precision medicine’s’ greatest contribution to the field, though current techniques may not be applicable. An alternative possibility is that asthmatics lie at one end of the normal distribution of airway sensitivity given that there is no clear cut off that can identify for certain that an individual will experience symptoms attributable to ASM shortening.

The frequency and severity of airway narrowing will, to a large degree, be influenced by sensitivity (how easily the airways narrow) and reactivity (how much they narrow), respectively [1]. An individual may have frequent symptoms but never have an attack severe enough to be hospitalised while another may have only two or three episodes in a year and yet be hospitalised each time. In contrast, healthy individuals without asthma can inhale very high doses of agents such as histamine without significantly narrowing their airways, presumably due to intrinsic counteracting homeostatic mechanisms. Inhaling β-agonists prior to inhaling histamine or participating in vigorous exercise can, in most asthmatics, restore control. The β-agonist causes both a mild dilation initially prior to the challenge and then effectively maintains stability of the airways through the challenge. Curiously, while β-receptors are plentiful and widely distributed throughout the airways, the airways do not receive sympathetic innervation. Importantly, they are subject to tachyphylaxis (de-sensitisation) when repeatedly stimulated and hence β-agonists alone cannot be used for long-term therapy without inhaled corticosteroids.

## 5. The One Reliable Physiological Read-Out Signifying Asthma

If the one unique feature of asthma is the tendency for ASM to constrict in response to positive stimuli due to failure of counteracting homeostatic mechanisms, then the pre-requisite for a reasonably confident diagnosis of asthma and inclusion in asthma studies must be the demonstration that a patient’s airway calibre will change rapidly in response to agents acting on ASM. One would not include a patient in a diabetic study without a physiological read out confirming that control of blood sugar is defective nor a study of anaemic patients without a blood count.

Sadly, this is rarely undertaken at a population level and, as noted before, ‘doctor-diagnosed asthma’ without objective data is hardly worth the paper it is written on, with an accuracy of approximately that of, or worse than, tossing a coin [17,18]. Similarly, questionnaire-based assessments of asthma are grossly flawed and able to generate very different prevalence figures depending on the questions used [17,19].

The most common ‘test’ is assessing the response to a short-acting β-agonist such as salbutamol or terbutaline, with a positive response being taken to be >12% improvement in the forced expiratory volume in 1 s (FEV1) or how much air can be exhaled in the first second of forcefully breathing out from full lung capacity [20,21,22] (though there is some debate regarding the most discriminating cut-off value). This simple test is unfortunately rarely undertaken outside specialist clinics and results are only valid if robust reproducibility criteria are met (to ensure the results are not down to changes in effort or the effect of practice). Of course, a positive result is only positive if the subject already has some ASM shortening and hence those at the moderate to mild end of the spectrum may have a negative test while a false-negative result can be also due to the loss of β-agonist responsiveness observed in viral exacerbations. Few pre-school children are able to master the technique, adding to the many challenges in establishing a robust diagnosis in this age group. Corticosteroids are able to shift the sensitivity of airways and hence reversibility is uncommon in those taking their medication as their baseline lung function in most will be close to normal [17,22,23]. Conversely, the stability of the airways can be challenged with pharmacological agents such as methacholine and histamine, specific allergens, as when investigating possible occupational asthma, or other less specific challenges such as exercise and the inhalation of hypertonic saline or mannitol powder [24,25]. These challenges are performed much less commonly than bronchodilator responsiveness and interpretation of the results can still be problematic. Other responses that provide similar information would be a complete and dramatic cessation of chronic symptoms after 5–8 weeks of inhaled steroids or after 5–7 day of oral steroids though clinicians rarely record these data.

Without a positive physiological confirmatory test, the diagnosis of asthma can at best be ‘possible asthma’ in the same manner that a person reporting passing lots of urine cannot receive a diagnosis of diabetes until a blood sugar result becomes available as there are other causes of passing what appears to be excessive amounts of urine. Conversely, many with type 2 diabetes have few if any symptoms at diagnosis. Similarly, pallor alone is not sufficient to include a patient in a study of patients with anaemia.

The importance of objectively confirming the diagnosis was demonstrated in Finland, where a confirmed diagnosis was associated with improved compliance, a huge fall in hospitalisation and almost elimination of asthma deaths [26,27]. In essence, doing the simple things works well!

## 6. Known Exacerbating Factors

### 6.1. Allergies

Many consider allergy and asthma as going hand in hand but the majority of those with allergies do not have asthma, while worldwide most asthmatics are not atopic [17,28,29]. What is clear is that allergy is neither sufficient nor necessary. As noted above, there are many with allergies to inhaled allergens who cough and produce increased secretions in response to exposure (have asthma-like symptoms) but who do not significantly narrow their airways and so do not have asthma but rather an allergic bronchitis, with the inflammatory response typically containing elevated eosinophil levels. The majority of those with asthma worldwide are not prone to allergies (non-atopic). If, however one has allergies affecting the airways and asthma, the mediators released in response to an allergen will cause the asthmatic airways to narrow and hence may trigger more symptoms than many non-allergic individuals. Thus, while there is no clear evidence that allergies can initiate asthma, allergic reactions can certainly induce symptoms and potentially make the condition more problematic. Similarly, an addiction to ice cream may significantly destabilise blood sugar control in diabetes but is not the cause of the diabetes.

### 6.2. Viral Infections

Viral respiratory infections undoubtedly cause ‘exacerbations’ though the mechanism is unclear. During an exacerbation, the normally rapid response to a β-agonist is lost, effectively locking down the airways to a variable degree from partial to severe [17,30,31,32]. In such circumstances, corticosteroids hasten recovery of β-agonist responsiveness. The cause for the loss of β-agonist responsiveness is not understood but viral RNA has been implicated in initiating the process in some studies.

In childhood, it is estimated that more than 90% of true asthma admissions result from an exacerbation initiated by a viral respiratory tract infection. During such an event, the airway inflammation may be the product of at least four processes (i) in response to the viral infection (predominantly neutrophil airway response), (ii) in response to an inhaled allergen such as house dust mite faeces in an allergic individual (typically elevated eosinophils), (iii) in response to airway narrowing which is likely to induce an inflammatory response from stressed epithelial cells, ASM, etc., and (iv) in response to a bacterial dysbiosis (chronic bacteria bronchitis) in those, typically poorly controlled asthmatics, with this as a comorbidity (typically neutrophilic) to which one might add the effects of inhaling tobacco smoke or environmental pollutants.

## 7. Most ‘Pre-School Wheeze’ Is Not Asthma but a ‘Snotty Lung’

As with allergies, patients can develop asthma-like symptoms without any significant closure of their airways due to ASM shortening. This is particularly common in the pre-school years, when symptomatic viral lower respiratory tract infections are most prominent. All symptomatic respiratory viral infections tend to cause some lower respiratory tract inflammation, manifesting as a cough and increased secretions. Cough receptors are said to exist below the vocal cords and hence a cough implies some lower respiratory tract involvement. In young children, mucosal oedema and particularly increased airway secretions can create obstruction and flow limitation. In effect, the child has a snotty nose very similar to the snotty nose which can cause significant difficulties for infants and young children. Flow limitation may result in the generation of a true wheeze (a musical sound that is most prominent on exhaling) [33,34]. However, the same process with mucosal oedema and airway secretions may also result in the generation of a ‘ruttle’ [35,36], course harsh inspiratory and expiratory noises, flowing through airways containing significant amounts of secretions. These are often erroneously called wheeze by clinicians, hence all that wheezes is not asthma and all that wheezes is not wheeze, particularly in, but not limited to, the pre-school years.

Wheezing in young pre-school children is far more commonly not attributable to asthma though some of those who wheeze with a virus in the second or third year of life will have an exacerbation of asthma [33]. Identifying those with significant ASM shortening is extremely challenging, the only reliable confirmation being a dramatic and unequivocal response to asthma medication. By school age, relatively few otherwise healthy children develop a wheeze in response to a viral infection and hence most but not all with genuine wheeze during a respiratory viral infection probably have asthma.

Scientists may read about ‘phenotypes of wheezing illness’ in early childhood. These reports should also be handled with the greatest of scepticism. A phenotype is an individual’s observable traits, whereas these so called ‘phenotypes’ are simply retrospective attribution of patterns of wheezing such as *‘early transient wheeze’*! [17,33,34]. None of them equate to asthma. In adult medicine, it can be difficult to distinguish patients with asthma, particularly in the older age group, not least because they do co-exist in some patients. Pre-school wheezing children provide an equivalent challenge for those looking after children.

If it looks like a duck, and quacks like a duck, we have at least to consider the possibility that we have a small aquatic bird of the family Anatidae on our hands *Dirk Gently’s Holistic Detective Agency. Adams D.* (but it is only a possibility)

The limited repertoire available to the airways in responding to environmental stimuli results in an extensive list of conditions that might appear to be due to asthma if one does not ensure that reversible airway narrowing due to ASM shortening is confirmed. Typical symptoms of asthma quoted in guidelines are cough wheeze and shortness of breath, all of which are common in other conditions; so while it might look like asthma, it frequently is not asthma or is asthma with a co-morbidity, with the co-morbidity being responsible for much of the on-going symptoms.

### All That Wheezes Is Not Asthma and All That Wheezes Is Not Wheeze

As noted above, all that wheezes is not wheeze and wheeze does not equate to asthma [35,36,37]. The old adage that all that wheezes is not asthma has been taught to medical students for many, many decades but is all too often forgotten or ignored while the reproducibility of reporting wheeze by clinicians and patients is poor [38,39]. Notes made in medical records regarding ‘added’ sounds such as wheeze should be treated with considerable skepticism.

Conversely, those with chronic airflow limitation (and hence poor lung function) may not wheeze; presumably due to alterations in respiratory mechanics. A wheeze is generated by vibration of central airways when more energy being applied through relaxation of chest wall muscles than can be translated into airflow due to the flow limitation and hence represented wasted energy in the same way that heat generated by friction dissipates some of the energy being used to move an object. With long term impaired function such as that seen in chronic severe asthma patients may not wheeze despite very and will often report they are’ fine’ having physiologically adapted to the chronic impairment. Hence, the absence of wheeze does not exclude asthma any more than wheeze signifies asthma.

## 8. Cough and Shortness of Breath—Same Rules Apply

Cough is the default response to an irritated lower airway, aimed at expelling potentially harmful material including sputum. The quality of the cough can, to an extent, help in trying to understand the nature of a cough, with a ‘wet cough’ signifying secretions in the airways. Asthmatics typically have a ‘dry’ cough, but it may be ‘wet’ during recovery from an exacerbation, as a result of untreated or poorly controlled symptoms and with a comorbidity such as a persistent bacterial bronchitis (bacterial dysbiosis), which is not uncommon in poorly controlled asthma. Parental reporting of the quality of a cough is unreliable for many reasons [40], not least amongst them being the variability of a child’s cough through the day and night.

Shortness of breath is a feature of many forms of respiratory and cardiac diseases as well as being common in those who are unfit, overweight and those who have dysfunctional breathing [41,42]. Without objective evidence of rapid reversibility (other than during a viral exacerbation), one cannot be confident that reported shortness of breath, be it with exercise, nocturnal or some other trigger, is related in any way to asthma. Dysfunctional breathing is probably more common amongst those with respiratory disease; but in many, there is no airway disease, and the condition represents a habituated response to chronic stress or specific stressful events.

## 9. Difficult Asthma

Publications frequently refer to ‘difficult asthma’. It is important to recognise that this is not the same as ‘severe’ asthma. Difficult asthma is defined as persistent symptoms despite taking high doses of conventional therapy. While many make this area more complicated than it might be, the causes of ‘difficult asthma’ are

It is not asthma and symptoms are due to other conditions,Asthma and one or more comorbidities with the comorbidities generating symptoms, andThe patient is not taking their prescribed medication or failing to take it correctly.

These are summarised in Figure 1.

Severe asthma that cannot be controlled with conventional treatment is very uncommon in children and this is probably the case in adults. It should be remembered that those who take their medication as prescribed generally have few problems and rarely bother their physician, while those who do not take their medication or have other causes for their symptoms are the ones that frequently return to their doctors (or just live with it having failed to find effective help).

## 10. Where Can Precision Medicine Contribute to Improved Outcomes in Asthma?

Most medical treatments are designed for the “average patient” as a one-size-fits-all-approach, which may be successful for some patients but not for others. Precision medicine, sometimes known as “personalised medicine”, is an innovative approach to tailoring disease prevention and treatment that takes into account differences in people’s genes, environments, and lifestyles. The goal of precision medicine is to target the right treatments to the right patients at the right time [43].

Asthma is a condition with a wide range of severities such that one patient may use a β-agonist a couple of times a year with colds and another is hospitalised many times a year (though causes of ‘difficult asthma’ outlined above, especially poor compliance, rather than ‘severe’ asthma commonly account for repeated presentations to hospital).

There is little evidence to convincingly suggest that the core defect is heterogeneous, though it maybe. It is important to understand that ASM hyper-reactivity lies at its heart, while taking into account the role of exacerbating factors such as allergic inflammation and respiratory viruses in manifestation of the symptoms as well as the contribution of co-morbidities to on-going symptoms that generate asthma-like symptoms.

## 11. Treatment

‘Personalised Medicine’ is not new, though. The best clinicians have always attempted to apply the most suitable treatments for an individual patient, taking into account not only what is known about a particular condition but also how the condition and treatments may be positive or harmful and impact on a particular condition and co-morbidities, all in the context of the patient’s wishes and beliefs. Clinicians have long focused on the patient’s environment and lifestyles and understood that there is a tendency for asthma to run in families [44]. Historically, a wide range of treatments, most of which were ineffectual or frankly dangerous, were used [45]. The medication which had a positive effect on airway tone used for the longest period was the anti-cholinergic thorn apple or *Datura stramonium*, which was used in asthma cigarettes for almost 200 years and was last used in the 1990s [46]. Inhaled adrenaline was found to be effective in the first half of the C20th and had a more rapid onset with fewer side effects. In the 1950sm it was the first treatment to be delivered by a pressurised metered dose inhaler, replaced a decade or so later with more selective β2 agonists such as salbutamol [46]. High doses of less selective agonists such as isoprenaline, and later fenoterol, did provide symptomatic relief but unfortunately were associated with increased mortality in younger subjects.

In the early 1970s, Dr Harry Morrow Brown [47] was able to demonstrate that inhaled beclomethasone was able to provide the benefits of oral steroids with a greatly improved safety profile, and inhaled steroids have since become standard therapy, working in both atopic and no-atopic individuals. While they do reduce eosinophilic inflammation, they critically alter airway sensitivity [23,48] (though probably having little effect on reactivity). Without treatment, bronchial sensitivity is stable over time [49] and certainly guidelines do not recommend giving ICS to non-atopic asthmatics. More recently, combined ICS and long-acting β-agonist [LABA] inhalers have been widely used for those who appear not to respond well to inhaled corticosteroids (ICS) alone (though much of this may be due to poor adherence, with ICS being administered when the patient has symptoms). More recently still, it is being argued that an ICS with fast-acting LABA used as needed may be the simplest approach to treating mild to moderate asthmatics [50]. This may be effective in milder cases by helping to prevent airways getting lock down when challenged by a virus, with a combination of bronchodilation and corticosteroids helping to maintain β-agonist sensitivity.

In reality, most patients with asthma and related conditions are looked after by primary care physicians who have no particular expertise in the area, with asthma as just one of a vast array of conditions they are expected to look after. Hence, a simplified one-size-fits all approach is as much as one might expect most to take from the detailed guidelines they are bombarded with on a daily basis. For most patients who take their medication regularly and effectively, this works well.

For children with difficult asthma and significant allergies, options include desensitisation (not widely practiced in the UK because it has long been felt to be dangerous in asthmatics, but more widely undertaken in some other countries) or the use of ‘biologics’ such as anti-IgE, anti-IL5 and anti-IL4RA monoclonal antibodies [51]. However, the number in whom such extreme measures are required if all other ‘treatable traits’ including comorbidities and non-compliance are addressed is a tiny proportion of asthmatic children.

## 12. Treatable Traits’ as Personalised Medicine?

A recent trend has been to try and move away from diagnostic labels and suggest that treatment of a patient with respiratory symptoms should be ‘personalised’ taking into account ‘treatable traits’ [52,53,54], that is treat each component that one can identify that is contributing to a patient’s problems. Examples include poor compliance and/or not using inhalers effectively [55,56] and co-morbidities such as a persistent bacterial bronchitis/chronic bronchitis with exacerbations (chest infections) [57,58], dysfunctional breathing [41,42] and conditions that impact on self-management such as depression. This is just good practice dressed up in new clothes! The ‘concept’ is no different to the teaching I received as a medical student more than 40 years ago, when we were told that when managing a chronic disease, we should consider the whole patient, their beliefs, those of their family and the possibility of other factors contributing to ongoing problems. Though this concept is being sold as ‘personalised medicine’, it is simply suggesting that a doctor do their job properly. If there is a condition that is causing symptoms irrespective of whether this is asthma or not, it should be treated appropriately—something all clinicians should be doing but all too frequently fail to do.

If the trend were to stop there, this would be fine, representing another attempt to encouraging clinicians to be more holistic and thoughtful in their care and encouraging consideration of alternative diagnoses or co-morbidities. Unfortunately, a more recent example appears to include some dangerous suggestions [54]. The authors argue that patients with eosinophilic inflammation should be treated with inhaled steroids and, if these fail, biologics such as anti-IgE or anti-IL-5 antibodies. Those who benefit from a β-agonist should be given a β-agonist and by implication if there is no evidence of airway eosinophilia, they should not be given corticosteroids. This reflects a belief that corticosteroids are simply anti-inflammatory and only act on eosinophilic inflammation in this setting, ignoring their critical impact on airway sensitivity [1]. The suggestion is all the more surprising given that one of the authors has argued that a stand-alone short-acting β-agonist inhaler should not be used to treat most asthmatics [59] and it has already been established that LABAs used without corticosteroids have been associated with increased mortality in some populations.

While addressing non-compliance, poor inhaler technique and identifying conditions that can cause asthma-like symptoms and need addressing in their own right represents good medicine and personalises treatment to an extent, it is not personalising the treatment of asthma per se.

## 13. Diagnosis

As noted above, the only robust diagnostic test is a dramatic and unequivocal response to asthma medication, ideally with objective lung function measurements. It has been suggested that biomarkers such a sputum or circulating eosinophils, markers of eosinophilic inflammation and exhaled nitric oxide levels amongst others may be helpful in diagnosing asthma but since they are all related to allergy or eosinophilic inflammation, it is not surprising that their sensitivity and specificity for asthma per se are poor [60,61,62] though they are useful in identifying atopic asthma (the adult onset non-atopic eosinophilic form of asthma is extremely rare in childhood if it exists at all). However, this currently carries little implication for management other than allergen avoidance where possible and, for some, desensitisation, but usually only those with evidence of a mono-allergy, whereas most atopic asthmatics are sensitised to multiple allergens.

## 14. Monitoring

Similar considerations apply to non-invasive monitoring with approaches such as exhaled nitric oxide (FeNO) and induce sputum quantification which have failed to reliability influence management even in those with severe atopic asthma [63,64]. Importantly, the stability of and correlation between markers such as sputum eosinophilia and FeNO are poor in children [65,66].

### 14.1. Genetics

Reviews of asthma from the C19th and earlier noted that a tendency to develop asthma tended to run in families but this certainly did not follow a Mendelian pattern. Over the past 30 years, vast sums of money have been spent on undertaking studies aimed at finding the key genetic risk factors for asthma and/or severity of asthma. These have generated numerous potential candidate genes but have so far failed to identify any ‘treatable traits’ based on an individual’s genetics. Moreover, there is no individual or clear cluster that consistently predicts risk and, to date, genotyping has not contributed to diagnosis [66].

Some data, not surprisingly, exist implicating certain polymorphism in a variety of genes with response to therapy but, as yet, these have not significantly impacted on disease management; but if resources were available, some of them may prove to be valuable, such as β-2 receptor polymorphisms [67,68].

Epigenetic changes have been investigated, again without clearly identifying any mechanism for the development of asthma or any treatable traits [69]; but given that asthma appears to be an acquired post-natal condition, this avenue may generate some valuable insights.

### 14.2. Environment and the Respiratory Microbiome

Many researchers have invested much in ‘early life’ events as the ‘cause’ for developing asthma. For many years, early respiratory syncytial virus (RSV) infections were touted as the cause for asthma based on apparent similarities in symptoms (cough, shortness of breath, and noisy breathing) until work showed that the outcome was much more likely to be attributable to host factors than the virus itself [70] (which infects all individuals). At the same time, many others were claiming early viral infections protected against asthma. More recently, interest in RSV as a cause of asthma has waned to be replaced by zealous promotion of rhinovirus (RhV) or subtypes of RhV such as RhVC as the ‘cause’ of asthma. Again, this is not supported by robust data but based more on extrapolating from the same similarities noted above and the frequency of RhV being identified as a trigger of exacerbations in older children. It should be noted that all infants experience multiple RhV infections, a significant number of which are asymptomatic, and therefore one would have to explain why the majority do not develop asthma. Equally relevant is noting that triggering an exacerbation (attack) of asthma is not the same as causing asthma in which symptoms commonly persist between infections. With hundreds of rhinovirus serotypes, re-infections are numerous. Being the most common symptomatic respiratory viral infection, it is not surprising it causes more exacerbations than other viruses.

Conversely, some have argued that early bacterial colonisation with the usual suspects of upper and lower airway infections predisposes to development of asthma, but all the same issues apply, not least being the failure to confirm that any reported symptoms at follow up are attributable to asthma. Much is still to be learnt about the role, if any, of the respiratory microbiome in the development of asthma. Asthma itself clearly impacts on the resident bacterial microbiome and, in many, particularly amongst those with poor control, the impaired mucociliary clearance predisposes to a chronic bacterial bronchitis that of itself generates significant symptoms [71,72].

### 14.3. Lifestyles

Asthma does appear to be more common in developed countries, but it is still unclear whether this is down to fashions in diagnosis, with over diagnosis in developed countries and underdiagnosis in developing countries, or whether this is a genuine difference. Studies based on questionnaires need to be treated with some scepticism for many of the reasons outlined above; and without objective assessment of reversibility, this remains speculative even in 2021. In the 1990s, it appeared that asthma was more common in countries such as West Germany and chronic bronchitis in less affluent, heavily industrialised countries such as East Germany. These differences appeared to rapidly disappear after reunification but again it is far from clear whether these were genuine differences or diagnostic transfer (changing the name of a condition). The hygiene hypothesis has been popular for many years, but the exact role of environmental factors in the causation of asthma is still far from clear, with much still to be explained [73]. If the postulated environmental factors do play a role, are they directly inducing undue reactivity of ASM or, as seems more likely, impacting on the development of allergic response, which in turn may induce symptoms in those with asthma?

### 14.4. Other Omics

Metabolomics and proteomics have been suggested as potentially having a role in diagnosis, monitoring or guiding therapy but, to date, this suggestion has not been borne out. Monitoring volatile agents in exhaled breath has been advocated by some [74,75] and some have used the so-called e-nose (which does not attempt to identify specific components of the exhaled breath) [76] but again, so far, without clear impacts on understanding of airway disease and clinical practice.

## 15. Summary

Given that clinicians cannot agree upon what they mean by asthma, this provides huge challenges for scientists attempting to find novel means of applying ‘personalised medicine’ to asthmatic patients. If we agree that the defining factor is ASM hyper-sensitivity and hence their ability to respond to both constricting and relaxing agents, this is a start. Co-morbidities are common and clear characterisation of patients is a must for any study, whether it be looking at basic mechanisms or trying to influence clinical outcomes.

As noted above, a hospitalised paediatric asthmatic patient is likely to have significant bronchoconstriction, with excessive secretions and clear evidence of an inflammatory response. Factors driving an inflammatory response will include the host response to the virus, any pre-existing allergic inflammation, cytokines released from stressed compressed airway cells, exacerbation of any bacterial dysbiosis and inflammation due to inhalation of toxin either in the form of cigarette smoke (passive in the home or active in older children) or from living in areas with high levels of environmental pollutants. Unravelling these intertwined stands is certainly a challenge and one that requires clear and rigorous thinking if we are to make significant progress.

## Figures and Tables

**Figure 1 jpm-12-00082-f001:**
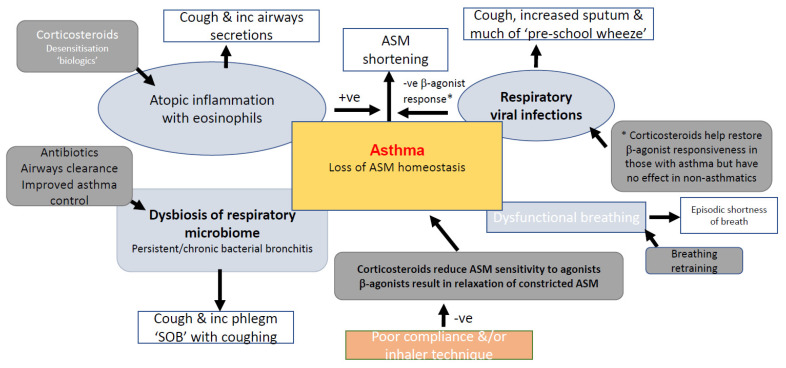
Conditions that can mimic features of asthma and/or influence the manifestation of asthma. +ve indicates increased effect; −ve indicates a negative effect.

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
