# Peer review of "Precision Medicine and Childhood Asthma: A Guide for the Unwary"

_jpm, 2022, doi:10.3390/jpm12010082_

Round 1
Reviewer 1 Report
This a comprehensive narrative review written by an expert in the field. The topics are well known for many pediatric pulmonologist, but also for this group there may be a need for emphasizing the main message. However, most of these children are treated by GPs for which this article may be long and overwhelming.
Moreover, the article is comprehensive in describing the challenges, but appears less as a Guide (as stated in the heading) on how to deal with the individual patient. Hence, diagnostic considerations could have been enhanced, especially how diagnostic possibilities (including history) may lead to a personalized approach to treatment.
Also , the role of bronchial hyperresponsiveness (and sensitivity vs reactivity) could possibly be enhanced. Opposite, It is less clear why discussion on “causes of asthma” under the heading of monitoring is so relevant in this setting.
Minor; there are some spelling errors, eg
- Line 238: has a snotty LUNG (?) very similar to the snotty nose..
- Line 350 – high doses ?
Author Response
I am grateful for the reviewer’s positive and helpful comments regarding the manuscript. I would make the following comments regarding the points they raise:-
I would agree absolutely that the manuscript is unlikely to be suitable for GPs and I would certainly have written it quite differently if this was the target audience. However, GPs are unlikely to read articles from this journal given the immense time pressures they are under to make sense of a host of conditions and there are many publications regarding this topic in journals aimed at a medical audience.
This manuscript was written with a view to alerting scientists engaging in this area that it may be much more challenging than at first it may appear. Those without a medical background would not commonly realise that asthma, while common, is not only poorly defined but also frequently mis-diagnosed. Precision Medicine bring very powerful techniques to bare on problems such as those encountered in respiratory medicine but if the patients included in a study have a variety of pathologies, then the old adage, rubbish in rubbish out, will become all too familiar in this area of research.
I believe it is vital that scientists drawn to working in this field understand the challenges and are prepared to challenge clinicians as to the suitability of the subjects being studied. To subject individuals to procedures for research without the likelihood of robust data being generated is not only a waste of time and money it is also unethical.
As this is the key aim of the manuscript, I have altered the introduction to make it clear who the intended audience is and why it is important for them to read the article.
Line 238 While snot generally refers to nasal discharge the point being made is that during viral infections exactly the same type of secretions are present in the lower airways with similar effects including partially obstructing the lumen. Bronchospasm is not the only mechanism for causing airways obstruction.
Line 353 High does less….. has been change to High doses of less….
Reviewer 2 Report
This review has given emphasis on the medical aspect of asthma and the difficulties around its accurate diagnosis. It contains adequate reference to previously reported biomarkers associated with asthma, which have been under investigation as potential treatments but there is very little about host cellular responses to asthma and current hypotheses on this topic.
- the review will be benefit by a more extensive report of asthma associated biomarkers and alterations of respiratory tract/blood cellular landscape seen in asthma suspected patients
Author Response
I am grateful for the reviewer’s positive and helpful comments regarding the manuscript. I would make the following comments regarding the points they raise:-
This manuscript was written with a view to alerting scientists engaging in this area that it may be much more challenging than at first it may appear. Those without a medical background would not commonly realise that asthma, while common, is not only poorly defined but also frequently mis-diagnosed. Precision Medicine bring very powerful techniques to bare on problems such as those encountered in respiratory medicine but if the patients included in a study have a variety of pathologies, then the old adage, rubbish in rubbish out, will become all too familiar in this area of research. The first paragraph of the introduction has been altered to make this clear
There are numerous publications regarding ‘host cellular responses’ though many concern the effects of co-morbidities such as allergic inflammation rather than asthma per se. As the reviewer notes there are many hypotheses regarding the origins and persistence of the condition, but they are just that, hypotheses. Before considering these tt is vital for scientists to be aware of the key components of the condition and the confusion caused by applying the term loosely or inaccurately.
I would agree that if this were a standalone manuscript it should refer to some of the many hypotheses regarding aetiology and persistence of the condition and to some novel biomarkers under investigation. However these will be covered in some detail in other manuscripts covered in this special edition on Precision Medicine in Childhood Asthma.
I trust these comments and minor changes to the manuscript fully address the issues raised by the referees and that manuscript is now suitable for publication.